# Effects of Normobaric Hypoxia and Adrenergic Blockade over 72 h on Cardiac Function in Rats

**DOI:** 10.3390/ijms241411417

**Published:** 2023-07-13

**Authors:** Elias Neubert, Beate Rassler, Annekathrin Hoschke, Coralie Raffort, Aida Salameh

**Affiliations:** 1Carl-Ludwig-Institute of Physiology, University of Leipzig, 04103 Leipzig, Germany; elias-neubert@gmx.de (E.N.); anne.hoschke@web.de (A.H.); 2Department of Pediatric Cardiology, Heart Centre, University of Leipzig, 04289 Leipzig, Germany; coralieraffort@orange.fr (C.R.); aida.salameh@medizin.uni-leipzig.de (A.S.)

**Keywords:** normobaric hypoxia, adrenergic blockade, cardiac function, hypoxia-inducible factor-1α, nitrotyrosine, apoptosis, poly-ADP-ribose, poly-ADP-ribose-polymerase 1

## Abstract

In rats, acute normobaric hypoxia depressed left ventricular (LV) inotropic function. After 24 h of hypoxic exposure, a slight recovery of LV function occurred. We speculated that prolonged hypoxia (72 h) would induce acclimatization and, hence, recovery of LV function. Moreover, we investigated biomarkers of nitrosative stress and apoptosis as possible causes of hypoxic LV depression. To elucidate the role of hypoxic sympathetic activation, we studied whether adrenergic blockade would further deteriorate the general state of the animals and their cardiac function. Ninety-four rats were exposed over 72 h either to normal room air (N) or to normobaric hypoxia (H). The rodents received infusion (0.1 mL/h) with 0.9% NaCl or with different adrenergic blockers. Despite clear signs of acclimatization to hypoxia, the LV depression continued persistently after 72 h of hypoxia. Immunohistochemical analyses revealed significant increases in markers of nitrosative stress, adenosine triphosphate deficiency and apoptosis in the myocardium, which could provide a possible explanation for the absence of LV function recovery. Adrenergic blockade had a slightly deteriorative effect on the hypoxic LV function compared to the hypoxic group with maintained sympathetic efficacy. These findings show that hypoxic sympathetic activation compensates, at least partially, for the compromised function in hypoxic conditions, therefore emphasizing its importance for hypoxia adaptation.

## 1. Introduction

A typical effect of ischemia or hypoxia on cardiac function is reduction in contractile force. In previous rat studies, we observed a significant decrease in left ventricular (LV) systolic pressure and contractility induced by acute normobaric hypoxia with 10% O_2_ [1,2]. This LV depression became significant after 6 h of hypoxia but showed a slight tendency towards recovery after 24 h of hypoxia. Impairment of LV systolic function in hypoxia has been confirmed in several animal studies. In rodents, acute hypoxia reduced mean arterial pressure, stroke volume and cardiac index and induced anaerobic metabolism, impairment of adenosine triphosphate (ATP) synthesis and mitochondrial damage in cardiomyocytes [3,4,5]. 

Limited O_2_ availability compromises the energy metabolism of the heart, thus reducing the capacity to synthetize ATP. However, the two ventricles are differentially affected by hypoxia. Experimental data from humans and rats showed that hypoxia significantly reduced oxidation of energetic substrates, mitochondrial oxidative phosphorylation and ATP synthesis in the LV [6,7,8], while the right ventricle (RV) was not affected [9]. These metabolic impairments were associated with depressed contractility and diastolic dysfunction in the LV [6,10]. Correspondingly, our previous studies showed that hypoxic depression was only present in the LV but not in the RV [1,2].

Moreover, impaired oxygenation of the myocardium leads to nitrosative stress and cell apoptosis or necrosis, conditions that may further reduce cardiac function. Typical reactions induced by hypoxia include formation of nitrotyrosine from nitrosylation of tyrosine residues by peroxynitrite radicals and activation of poly-ADP-ribose polymerase-1 (PARP-1) [11]. PARP-1 depletes cellular energy and mediates cell death via upregulation of poly-ADP-ribose (PAR) and translocation of apoptosis-inducing factor (AIF) from the mitochondria to the nucleus [12]. 

Exposure to hypoxia induces a number of reactions that help the organism to cope with this challenge. The transcription factor hypoxia-inducible factor (HIF)-1 is a key regulator of adaptation to hypoxia. It is involved in erythropoiesis, angiogenesis, glucose and energy metabolism, cell proliferation and viability and many more adaptation reactions. In the heart, HIF-1 plays an important cardioprotective role [13] that may contribute to restore the depressed LV pump function. Moreover, hypoxia is associated with activation of the sympathetic nervous system, which is mediated by stimulation of peripheral chemoreceptors [14,15,16]. Sympathetic activation may compensate for the hypoxic depression of LV function to a certain extent [17,18]. In our previous study on rats exposed to normobaric hypoxia, combined α- and β-adrenergic blockade (as a model of a lack of hypoxic sympathetic activation) further deteriorated LV function and prevented the trend towards recovery after 24 h. However, hypoxia and additional norepinephrine (NE) infusion did not improve cardiac function. On the contrary, these animals suffered the same drop in left ventricular systolic pressure, heart rate and cardiac index as hypoxic control rats without NE application, and a large number of them died prematurely with signs of acute RV failure and severe pulmonary edema [1]. These results indicate that both a lack of sympathetic activation and sympathetic overactivation can impair cardiopulmonary adaptation to hypoxia. Increased levels of catecholamines or adrenergic overstimulation may exert cardiotoxic effects [19]. Particularly, β-adrenergic hyperstimulation may induce myocyte damage, cardiac dysfunction and finally heart failure [20,21]. Thus, hypoxic sympathetic activation may combine positive (sufficient oxygen supply to the tissues) and negative effects (potential cardiac damage), and the relation between them may be of high importance for people under hypoxic conditions.

This study was designed to examine the effects of prolonged exposure to normobaric hypoxia over 72 h on rats. The main question was whether adaptation to hypoxia is detectable after 72 h and may be associated with recovery of cardiac function. The second question was directed to the possible reasons of hypoxic LV depression. Specifically, we hypothesized that nitrosative stress, ATP deficiency and apoptosis are important reasons for the hypoxia-induced reduction in LV pump function. Finally, we were interested in the role of hypoxic sympathetic activation. For this reason, we studied whether adrenergic blockade would further deteriorate the general state of the animals and their cardiac function.

Markers of the nutritional condition, metabolism and blood gases were determined to assess the general state of the animals. Hemoglobin (Hb) concentration, hematocrit and the expression of HIF-1α served as markers of adaptation. LV and RV catheterization provided parameters of cardiac function. We determined the expression of nitrotyrosine, PARP-1, PAR and AIF as markers of nitrosative stress, ATP deficiency and apoptosis, respectively.

## 2. Results

### 2.1. Nutritional Condition and Blood Analysis

Animals in the hypoxic cohort tolerated the experiment less well than normoxic animals. Hypoxic rats took up significantly less food (42% compared to normoxic animals; *p* < 0.001) and showed a significant loss in body weight (BW) by about 12% of their baseline weight (*p* < 0.001 compared to the normoxic cohort). Moreover, the hypoxic cohort had a significantly higher blood glucose concentration than the normoxic cohort (*p* < 0.001). Within the cohorts, application of adrenergic blockers did not induce significant differences between the groups (Table 1).

Hemoglobin concentration and hematocrit were significantly higher in the hypoxic cohort than in the normoxic cohort, but arterial oxygen saturation (S_a_O_2_) was significantly reduced (*p* < 0.001). Application of adrenergic blockers had no significant effect in normoxic animals. Only in the H-PR group, S_a_O_2_ was significantly higher than in the hypoxic control group (Table 2). Arterial pH and partial pressure of carbon dioxide (pCO_2_) were also significantly lower in hypoxic animals. These values were associated with a negative base excess (BE; −10.4 on average) indicating a partially compensated metabolic acidosis. Notably, in the hypoxic groups with adrenergic blockers, BE values were significantly less negative than in the hypoxic control group. In the normoxic cohort, all three parameters were in the normal range (Table 2). 

### 2.2. Complications Prior to or during Hemodynamic Measurements

After administration of thiopental narcosis for heart catheterization and during the course of the hemodynamic measurements, 22 animals from the hypoxic cohort (39%) suffered cardiorespiratory arrest one or more times and needed reanimation. After short-term exposure to normoxia (1–3 min), 16 animals were successfully reanimated. Only six animals (two each from H-Ctrl and H-PR groups and one each from H-PZ and H-PZ+PR groups) died prior to the completion of hemodynamic measurements with symptoms of LV decompensation and acute RV failure, such as a dilated right heart and congestion in the lungs and the liver. From the normoxic cohort, four animals (two each from N-PZ and N-PR groups) died during hemodynamic measurements with similar symptoms. 

### 2.3. Results of Hemodynamic Measurements

The most important hemodynamic results are presented in Figure 1 and Table 3. Hypoxia induced a significant deterioration in the pump function of the LV. Stroke volume (SV; Figure 1), ejection fraction (EF; Table 3), stroke work (SW; Table 3) and cardiac index (CI; Figure 1) were significantly reduced in the hypoxic compared to the normoxic cohort. Similarly, hypoxic control animals (H-Ctrl group) showed a significant reduction in systolic inotropic function in comparison to the N-Ctrl group as reflected by LV systolic pressure (LVSP) and contractility (LV dP/dt max) (Figure 1). Application of adrenergic blockers reduced LVSP even in normoxia as compared to N-Ctrl animals. With α-adrenergic blockade, however, LV contractility, SV, EF, SW and CI remained at normoxic control level. In contrast, β-blockers (alone or in combination with PZ) significantly decreased these parameters by about 20–30% compared to normoxic control. In hypoxia, β-adrenergic blockade (PR or PZ+PR) induced only slight further decrease in these parameters, while α-adrenergic blockade reduced them to a similar level to the β-blocker groups. The effects of hypoxia and adrenergic blockade on heart rate (HR; Table 3) were similar but reached a lesser extent. Similar to contractility, LV relaxation (LV dP/dt min; Figure 1) was affected both by hypoxia and by adrenergic blockade. In particular, β-adrenergic blockade (PR or PZ+PR) decreased LV dP/dt min by 30–40% in normoxia and hypoxia compared to the corresponding controls. Impaired relaxation and reduced cardiac output resulted in an elevated end-diastolic pressure (LV edP; Figure 1). This effect was most pronounced in the H-PZ+PR group. Moreover, although LV dP/dt min of these animals was significantly lower than that of normoxic control rats, they showed a similar LV end-diastolic volume (LV edV; Figure 1) to that of normoxic controls. However, end-systolic elastance (LV esE; Table 3) was not significantly elevated in hypoxia.

While LV function was significantly impaired by hypoxia, systemic circulation was not compromised. There were no significant differences in mean aortic pressure (MAP; Table 3) between the normoxic and hypoxic cohorts (96.4 ± 3.5 and 91.0 ± 2.4, respectively; *p* = 0.185). Total peripheral resistance (TPR; Table 3) was not affected by hypoxia alone, but the combination of hypoxia and β-adrenergic blockade (H-PR and H-PZ+PR groups) induced a mild increase in TPR compared to the H-Ctrl group, which was not present in the corresponding normoxic groups.

In contrast to LV function, RV function was not reduced by hypoxia alone. In the hypoxic cohort, RV systolic pressure (RVSP) was significantly elevated (*p* = 0.02; Figure 1). RVSP and RV dP/dt max were also slightly higher in the hypoxic than in the normoxic control group (Figure 1 and Table 3). In both conditions, adrenergic blockade gradually decreased RV function compared to the respective control groups, mainly with combined adrenergic blockade (PZ+PR). The results indicate that blockade of β-adrenergic effects affects RV function more than hypoxia.

### 2.4. Results of Immunohistochemical Analyzes

As to be expected, the expression of HIF-1α was in the hypoxic cohort more than twice as high as in normoxic animals (*p* < 0.001). In both cohorts, HIF-1α was higher in the RV than in the LV and reached in the hypoxic cohort about 50% of the total evaluated RV area. However, the relative increase from normoxia to hypoxia was greater in the LV (218%) than in the RV (179%). Addition of adrenergic blockers had only minor effects. In the RV, HIF-1α expression gradually increased in hypoxia with PZ, PR or PZ+PR, but these changes were not significant. In the LV, H-PZ+PR induced a significant HIF-1α increase compared to the hypoxic control (Figure 2).

Hypoxia induced significant nitrosative stress in the heart (normoxic vs. hypoxic cohort: *p* < 0.001), in particular in the RV. While NT increased under hypoxic compared to normoxic conditions in the LV by only 35%, it more than doubled in the RV. Interestingly, adrenergic blockade gradually increased the NT expression in the LV even in normoxia. In the N-PZ+PR group, NT values reached a similar level to the hypoxic control group. In hypoxia, NT values further increased with adrenergic blockade, most of all with H-PZ+PR. In contrast, adrenergic blockade had only marginal effects on NT in the RV in both normoxia and hypoxia as compared with the corresponding controls (Figure 3).

Hypoxia was accompanied by significant activation of PARP-1, upregulation of PAR and nuclear translocation of AIF. For all three markers, the relative increase in hypoxia (related to normoxia) was similar in both ventricles (normoxic vs. hypoxic cohort: *p* < 0.001). PARP-1 increased to more than three-fold values and induced a more than two-fold increase in PAR. Hypoxia plus singular adrenergic blockade decreased the levels of PARP-1 and PAR in both ventricles compared with hypoxic controls, in particular with PR. In contrast, with combined adrenergic blockade PARP-1 and PAR further increased to higher levels than in hypoxic controls (Figure 4 and Figure 5).

Correspondingly, the number of AIF-positive nuclei increased in hypoxic compared to normoxic animals about 4.5-fold, indicating induction of apoptosis. AIF increase was similar in both ventricles with a slight predominance in the LV. The effects of adrenergic blockers on AIF were similar to those of PARP-1 and PAR but did not exceed the values of the hypoxic controls (Figure 6). 

Hypoxia had no effect on relative heart weight (HW/BW, medians [25th; 75th percentiles]: N-Ctrl 3.21 [3.09; 3.37]; H-Ctrl 3.16 [2.91; 3.48]; *p* = 1.0). With adrenergic blockers, HW/BW decreased slightly both in normoxia and hypoxia. This reduction was only significant in the N-PZ+PR group (2.76 [2.68; 3.00]; *p* = 0.015 vs. N-Ctrl). 

## 3. Discussion

The results of the present study clearly showed no recovery of LV inotropic function between 24 h and 72 h of exposure to normobaric hypoxia with 10% O_2_ in N_2_. A previous study on rats exposed to the same hypoxic conditions over up to 24 h revealed that LVSP and LV dP/dt max decreased by almost 25% and even 50% of normoxic values, respectively, after 6 h of hypoxia. By 24 h, both values re-increased to 83% and 73%, respectively [1]. After 72 h of exposure, both values remained in a similar range with 84% and 80%, respectively. In addition, the present results showed that the reduction in pumping force and contractility was accompanied by a significant reduction in stroke volume by 14% and stroke work by even more than 30%. As HR was also reduced under hypoxic conditions, cardiac index significantly decreased by about 20% compared to normoxia. Hypoxia also affected LV relaxation, but this impairment was rather slight and not significant. These results confirm and corroborate the findings from our previous study [1]. In contrast, RV parameters were not compromised by hypoxia, thus confirming echocardiographic findings in humans under normobaric hypoxia conditions [22]. In the hypoxic animal cohort of the present study, RVSP was even higher than in normoxic animals. This may be attributed to increased capillary pressure in the lungs resulting from hypoxic pulmonary vasoconstriction, which is not confined to pulmonary arterioles but also has a considerable venous component [23]. In rats, the hypoxic vasoconstrictor effect on pulmonary veins was even significantly greater than on arteries [24]. Elevated capillary pressure in the pulmonary vascular bed requires the RV to increase its systolic pressure and contractility to overcome the increased vascular resistance. These findings are also in line with previous observations after shorter exposure to hypoxia [1]. Moreover, hypoxia exerts differential effects on adrenoceptors in the LV and RV, which may also account for the different responses of the LV and RV to hypoxia. Hypoxic stimulation modulated functional activity of G proteins by increasing Gi and reducing Gs activity and increased the density of α_1_-adrenoceptors in the LV but not in the RV [25].

In general, the adverse effects of hypoxia on LV function were greater than those of additional adrenergic blockade. We assume that hypoxic sympathetic activation in rats may be less effective than in humans. In contrast to our results in rats, studies in humans showed enhanced myocardial contractility, increased heart rate and unaltered stroke volume, resulting in improved cardiac output under hypoxic conditions [17,26,27]. In rats exposed to normobaric hypoxia, we observed only mild increases in plasma concentrations of epinephrine and norepinephrine, indicating that their hypoxic sympathetic activation was rather weak. Even infusion with norepinephrine could only delay but not completely prevent the LV depression in hypoxia [1]. In addition, hypoxia increases the clearance of norepinephrine [28], which might further attenuate the effects of hypoxic sympathetic activation. 

In normoxia, α-adrenergic blockade hardly affected the inotropic and chronotropic function of the LV. The reduction in LVSP may be attributed to a vasodilatory effect of α-adrenergic blockade resulting in a slight reduction in TPR and, hence, in the resistive load to the LV. RVSP was also slightly decreased by PZ administration. This may be explained in analogy to LVSP reduction as pulmonary arteries constrict in response to α-adrenergic stimulation [29,30]. Under hypoxic conditions, LVSP and RVSP did not further decrease. Stroke volume, HR and consequently CI significantly decreased compared to the N-PZ group. 

As expected, β-adrenergic blockade significantly decreased LV inotropic function, stroke work and cardiac output even in normoxia. In particular, the reduction in cardiac output may be reinforced by an increase in TPR resulting from the blockade of vascular β-adrenoceptors. The effects of PR were even more pronounced with combined α- and β-adrenergic blockade. β-Adrenergic blockade (alone and even more in combination with the α-blocker PZ) also compromised LV relaxation, but LV end-diastolic pressure and volume remained low under normoxic conditions. Hypoxia further deteriorated LV function, but this effect was rather small compared to the corresponding normoxic groups. RVSP was not compromised at all; only RV dP/dt max was reduced in the hypoxic compared to the normoxic PR group. Notably, in analogy to effects of hypoxia, adrenergic stimulation exerts differential effects on LV and RV, but the responses to hypoxia and to adrenergic stimulation differ from each other [25]. In general, the effects of single PR and of combined PZ+PR administration were similar under hypoxic conditions and were greater than effects of single PZ infusion. However, the most severe effect on LV inotropic and lusitropic functions was exerted by hypoxia and dual adrenergic blockade. With this treatment, LV dP/dt min reached its lowest and LV edP its highest values compared to all other groups. While LV pump function and relaxation were significantly reduced, LV edV was in the same range as in the normoxic control group. The slight improvement of LVSP, SV and EF in comparison to the H-PR group might be explained by the effect of the Frank–Starling mechanism: increased end-diastolic filling of the LV enhances contraction and increases stroke volume and EF [31,32]. The mildly elevated values of LV edV and RVSP in this group indicate an imbalance between RV and LV pump functions that may cause a backlog in the pulmonary circulation [1,2,33]. In our previous study [1], 11 of 12 rats exposed to hypoxia with PZ+PR infusion over 24 h presented with symptoms of acute RV failure at necropsy, with 10 of them having died prematurely during final thiopental narcosis. These symptoms were associated with signs of lung congestion, thus indicating an insufficient compensation for the depression of LV inotropic function, which was induced by hypoxia plus adrenergic blockade. We assumed that additional stress due to narcosis and heart catheterization finally resulted in decompensation and acute RV failure [1]. The lower incidence of premature deaths in the present study (11% of all hypoxic animals) indicates that the situation of the animals had stabilized after 3 days of hypoxia. Nevertheless, addition of adrenergic blockers means a great challenge to the organism as is reflected in the fact that four normoxic animals with adrenergic blockade in the present study died during thiopental narcosis. Combination of hypoxia with adrenergic blockade and other adverse factors such as narcosis or stress due to heart catheterization may induce a vulnerable condition of the animals that not infrequently proves fatal.

### 3.1. Adaptation to Hypoxia

Our immunohistochemical studies showed that hypoxia resulted in a significant increase in cytoplasmic HIF-1α expression. HIF-1 is a heterodimeric protein consisting of HIF-1α and HIF-1β subunits, with the α subunit being rapidly degraded under normoxic conditions. In hypoxia, however, HIF-1α is stabilized, accumulates in the cytoplasm and then forms together with HIF-1β the functional active transcription factor HIF-1. HIF-1 migrates into the cell nucleus and activates, among other things, the erythropoietin (EPO) gene with the aim of increasing the number of red blood cells to ensure sufficient oxygen supply to the tissue [34]. This is exactly what we observed in our study: all rats in hypoxia, both with or without adrenergic blockade, had significantly increased hemoglobin concentration and thus also elevated hematocrit values. Although we did not measure EPO plasma concentrations, we can assume with certainty that blood EPO levels were elevated in hypoxic rats. Moreover, it has been shown that in addition to the erythrocyte-stimulating function, EPO also exerts significant pleiotropic effects in the heart via EPO receptors. These receptors are found not only on cardiac endothelial cells and fibroblasts but have also been detected on cardiomyocytes [35]. Thus, it is conceivable that increased EPO production not only increases hemoglobin levels for better oxygen supply but also leads to an improvement in cardiac performance. In the heart its cardioprotective influence has already been demonstrated. Infarction studies (in vivo model) and ischemia/reperfusion studies (Langendorff model) revealed that EPO application led to a significant reduction in apoptosis and to an improvement of left ventricular function [35,36]. The HIF-1α/EPO pathway might therefore be an effective compensatory mechanism for hypoxia-induced cardiac damage. This is confirmed by the observation that only 11% of all hypoxic animals of the present study (four with and two without adrenergic blockade) died prematurely with signs of LV decompensation and acute RV failure, while such deaths occurred in 83% of rats exposed to hypoxia for 24 h [1].

In our study, the application of either PR or PZ had no significant impact on HIF-1α production. Only adrenergic blockade with PR+PZ during hypoxia induced a slight but significant increase in HIF-1α in the left but not the right ventricle. This slight increase in HIF-1α does not appear to have any physiological effect on hemoglobin and hematocrit levels and, consequently, on EPO production. Remarkably, the hearts with dual sympathetic blockade had the worst cardiac performance in both normoxia and in hypoxia. Upregulation of HIF-1α mRNA and protein expression has been observed in a rat model with experimental volume-overload heart failure [37]. We assume that the animals in the H-PZ+PR group might have been at the edge of transition into LV decompensation, which may account for this HIF-1α increase.

### 3.2. What Might Account for the LV Depression in Hypoxia?

Our results indicate that acclimatization to hypoxia has improved after 72 h of hypoxic exposure and stabilized the condition of the animals compared to 24 h of hypoxia. Nevertheless, even those animals surviving 72 h of exposure to hypoxia and the total experimental procedure presented with poor general condition: they ate less than half that of the animals remaining in normoxic conditions and, consequently, they lost body weight compared to the corresponding normoxic groups. Moreover, the hypoxic animals showed signs of metabolic acidosis, which may account for insufficient energy metabolism of these animals. These findings demonstrate that even after 72 h acclimatization to hypoxia must still be regarded as incomplete. While hemoglobin and hematocrit values in our normoxic cohort were in the same range as typical for rats raised at sea level, rats raised at 3600 m of altitude presented clearly higher values, with 20.4 ± 0.3 g/mL and 60.5 ± 1.1%, respectively [38], than our hypoxic animals. 

Compromised mitochondrial function and ATP synthesis have been demonstrated in hypoxic cardiomyocytes [6,8], indicating that cardiac energy deficiency may have contributed to the depression of LV function in hypoxic rats. Under hypoxic conditions, the heart switches its substrate preference for energy metabolism from fatty acid as the preferred fuel towards glucose [39]. The insulin-independent glucose transporter isoform GLUT1, which mediates the basal glucose transport into cardiac myocytes, is also activated by HIF-1 and is upregulated in the LV under hypoxic conditions [40,41,42]. Elevated blood glucose levels observed in the hypoxic animals of the present study indicate that this step in hypoxic adaptation might have been insufficient after 72 h of hypoxia. Reduced glucose availability in the cardiomyocyte may have impaired cardiac energy metabolism and, consequently, reduced myocardial pump function.

Chronic hypoxia can lead to nitrosative stress by induction of the inducible NO-synthase (iNOS), resulting in an increased production of peroxynitrite [43]. Peroxynitrite, which can be detected by an increase in nitrosylated tyrosine residues (nitrotyrosine), causes DNA strand breaks that in turn entail DNA repair mechanisms. The enzyme poly-ADP-ribose polymerase (PARP) restores DNA integrity in an ATP-consuming process thereby releasing poly-ADP-ribose (PAR) [44]. PAR in turn leads to the release of apoptosis-inducing factor (AIF) from mitochondria [45]. Hence, excessive ATP consumption occurs as a result of PARP activation and impairs the energy metabolism of the cardiomyocyte. Moreover, cell death is initiated by AIF release [46]. In our study, we showed that hypoxic conditions enhance nitrosative stress as reflected by increased NT levels within the heart. Furthermore, increased PARP levels and concomitant increases in the levels of PAR and AIF were shown. These findings demonstrate that hypoxia causes a deterioration of energy metabolism in the cardiomyocytes not only via a presumably reduced glucose supply but also PARP activation. In addition, hypoxia activates apoptotic signaling pathways. Taken together, these reactions to hypoxia resulted in a marked deterioration of left ventricular function. 

Studies with isolated cardiomyocytes showed that stimulation of adrenergic receptors induced different responses relating to apoptosis. While stimulation of α-adrenoceptors, in particular of α_1A_-adrenoceptors, had an anti-apoptotic effect [47], stimulation of β-adrenoceptors showed a marked increase in cellular apoptosis. The increase in apoptotic cells was caused by a ROS-dependent process with release of cytochrome C from mitochondria [48,49,50]. In rodents, β_1_- and α_1B_-adrenoceptors are the most prevalent adrenoceptors in ventricular cardiomyocytes, while α_1A_-adrenoceptors showed high levels in only 20% of ventricular myocytes [51]. This may explain why single α- or β-adrenergic blockade slightly reduced AIF release in normoxia as well as in hypoxia while AIF levels in dual adrenergic blockade reached similar levels to the control groups.

To summarize, the hypoxia-induced depression of LV function continued unabated after 72 h of hypoxia even though the animals showed clear signs of acclimatization to hypoxia. The significant increase in markers of nitrosative stress, ATP deficiency and apoptosis in the myocardium provides a possible explanation for the absence of the LV function recovery. Application of adrenergic blockers during exposure to hypoxia further deteriorated LV function. Even though this effect was rather low compared to the effect of hypoxia, the combination of hypoxia and adrenergic blockade may create a vulnerable condition for the organism. If combined with additional stressors or diseases, this vulnerable condition may easily result in fatal consequences. This illustrates the importance of hypoxic sympathetic activation for adaptation to hypoxia. If sympathetic activity or cardiac sympathetic innervation is reduced or even abolished in a hypoxic or hypoxemic condition, the risk for cardiac decompensation is high. This may have important implications for patients with multimorbidity, e.g., patients with both pulmonary and cardiovascular diseases, and especially for their treatment, e.g., with β-blockers. Our results show, in contrast, that hypoxic sympathetic activation can at least partially compensate for the compromised cardiac function under hypoxia conditions.

### 3.3. Limitations of the Study

At the beginning of the experiments, we had no equipment for blood gas analysis including measurement of Hct and Hb concentration. This possibility opened up later, but, initially, the blood gas analyzer was in a remote room. Therefore, the time intervals between withdrawal and analysis of the blood samples were often long, and this affected the analysis, resulting in many measurement failures. Later, the device was transferred into our laboratory, which considerably reduced the number of measurement failures. The original study design only comprised five animal groups (no normoxic blocker groups). Due to the possibility of blood analysis, we decided to expand the number of groups to the final study design including eight animal groups.

Another limitation was due to measurement failures in blood gas analysis and heart catheterization. If animals died prior to the planned exsanguination, the blood was often clotted before analysis. Moreover, despite the utmost caution during blood collection, blood samples sometimes contained small air bubbles. In these samples, the measurement of the partial pressure of O_2_ (pO_2_) failed. Finally, LV catheterization did not provide reliable volume data in some animals due to difficulties in adjusting the optimal catheter position. We compensated for such measurement failures by including more animals. 

## 4. Materials and Methods

### 4.1. Animal Model

All experiments were performed on 94 female Sprague-Dawley rats supplied by Charles River (Sulzfeld, Germany). The body weight was 236 ± 1.3 g at the beginning of the study corresponding to an age of about 10–12 weeks. All animal protocols were approved by the state agency (Landesdirektion Sachsen, number and date of approval: TVV 46/18; 17 December 2018). The experiments were conducted in accordance with the Guide for the Care and Use of Laboratory Animals published by the National Institutes of Health and with the “European Convention for the Protection of Vertebrate Animals used for Experimental and other Scientific Purposes” (Council of Europe No 123, Strasbourg 1985).

### 4.2. Study Protocol

Animals were subdivided into two cohorts to be exposed to normoxia (N) or normobaric hypoxia (H) for 72 h (Table 4). All animals received an intravenous infusion over the total experimental time. Infusions were administered with automatic pumps (Infors AG, Basel, Switzerland) at a rate of 0.1 mL h^−1^ via an infusion catheter (Vygon, Aachen, Germany), which was inserted into the left jugular vein. After catheter insertion, the animals woke up and moved freely with access to tap water and rat chow diet (Altromin C100, Altromin GmbH, Lage, Germany). Normoxic animals remained under room air conditions. Animals exposed to hypoxia were placed into a hypoxic chamber sized 65 × 105 × 50 cm^3^. The gas mixture in the chamber contained 10% oxygen in nitrogen. Special equipment prevented penetration of ambient air during manipulations on the animals, thus keeping the oxygen concentration in the chamber stable at 10 ± 0.5%. Exposure to hypoxic environment started immediately after catheter insertion. The normoxic and hypoxic cohorts were subdivided into 4 groups each. From both cohorts, one group was infused with 0.9% sodium chloride (NaCl) solution and served as normoxic or hypoxic control (N-Ctrl, H-Ctrl). The remaining groups (one group each from the normoxic and hypoxic cohort) were infused with the α-adrenergic blocker prazosin (PZ, 0.1 mg kg^−1^ h^−1^), the β-adrenergic blocker propranolol (PR, 0.16 mg kg^−1^ h^−1^) or a combination of the two (PZ+PR, 0.1 + 0.16 mg kg^−1^ h^−1^, respectively). These drugs are in clinical use in patients; this means that the use of these drugs may serve as a model for patients in a hypoxic condition, e.g., due to pulmonary diseases, in treatment with adrenergic blockers. For instance, β-blockers are administered to patients with coronary heart disease or after myocardial infarction. Among the β-blockers, we chose propranolol as it blocks both β_1_- and β_2_-adrenoceptors, thus inducing both cardiac and vascular effects. Prazosin is a blocker of α_1_-adrenoceptors that is applied in the treatment of arterial hypertension and of arterial ischemic conditions. Sympathetic vasoconstrictor effects are mainly mediated via stimulation of α_1_-adrenoceptors. 

The minimum number of animals within each group was defined as *n* = 8. Prior to the main experiments, we performed pre-tests to test the experimental and measurement conditions and the drug doses (*n* = 24). These pre-tests were conducted in the same way as the main experiments and included hemodynamic measurements but not feed and weight balancing and blood gas analysis. Initially, the design of the study only comprised 5 groups (i.e., 40 animals): a normoxic and a hypoxic control group as well as 3 hypoxic groups with adrenergic blockers. We always had 2 animals in the experiment at the same time receiving the same treatment. Assignment of the pairs to the 5 groups was performed using random numbers, which were created using the random number generator of Microsoft^®^ Excel. After the first part of these experiments has been conducted, we decided to add 3 groups with normoxic animals receiving adrenergic blockers to allow a direct comparison between normoxic and hypoxic animals with the same blocker treatment. Again, random numbers were used to allocate the remaining pairs of animals to the normoxic and hypoxic cohorts and to the different groups. Due to complications in the experimental course and measurement failures (see Section 3.3), we often obtained incomplete datasets. Therefore, we increased the number of animals per group until there were at least 6–8 values per group for each measurement. In addition, we included 4 normoxic and 8 hypoxic control animals from the pre-tests into the final evaluation.

### 4.3. Hemodynamic Measurements

About 30–40 min before the end of the exposure time, the animals were anesthetized with thiopental (Trapanal^®^ 80 mg kg 1, i.p.). After weighing, the animals were tracheotomized, and a polyethylene cannula was placed in the trachea. The right ventricle (RV) was catheterized with Millar^®^ (Millar Instruments, Houston, TX, USA) ultraminiature catheter pressure transducers. For catheterization of the left ventricle (LV) and recording of pressure–volume loops, a pressure–volume Millar catheter (Millar Instruments, Houston, TX, USA) was inserted into the left ventricle via the right carotid artery. The catheter was connected to a signal amplifier system (MPVS Ultra, Millar Instruments, Houston, TX, USA). Data acquisition and analysis were performed with the Power Lab 16/35 and Lab Chart Pro Software (Lab Chart 8) from ADInstruments (sales department FMI Föhr Medical Instruments GmbH, Seeheim, Germany). Parallel conductance was corrected by injection of isotonic saline solution (0.1 mL, 0.9% NaCl) and calibration of stroke volume by the thermodilution method (for more details see [52]). Thermodilution was performed for measurement of cardiac output using a thermosensitive 1.5F microprobe and a Cardiomax II computer (Columbus Instruments, Columbus, OH, USA). We measured LV and RV systolic pressures (LVSP, RVSP) and heart rate (HR). In addition, LV and RV maximal velocities of increase (dP/dt max) and decrease in pressure (dP/dt min) were determined as measures of ventricular contractility and relaxation, respectively. From LV catheterization, stroke volume (SV), ejection fraction (EF), stroke work (SW), end-diastolic pressure and volume (edP, edV) and end-systolic elastance (esE) were determined. After withdrawal of the LV catheter tip into the aorta, diastolic aortic pressure (DAP) was measured to calculate mean aortic pressure (MAP). Cardiac index (CI) was calculated as body mass-related cardiac output, and the total peripheral resistance (TPR) was calculated by dividing MAP by CI. Hypoxic animals remained in hypoxia until completion of hemodynamic measurements.

### 4.4. Sampling of Materials

After the hemodynamic measurements, the abdominal cavity was opened by midline incision. Animals were sacrificed by drawing blood from the abdominal aorta. A small sample of aortic blood was immediately added to a blood gas analyzer ABL800 BASIC (Radiometer Medical ApS, Brønshøj, Denmark) for oximetry and blood gas assessment. Blood glucose concentration was measured using the blood glucose monitor BGStar (AgaMatrix, Inc., Salem, NH, USA). After opening of the thoracic wall, the heart was excised and weighed, and the cardiac apex was fixated in formalin for immunohistochemical analysis. Finally, feed residues were weighed to determine the total feed intake of the animals. 

### 4.5. Immunohistochemistry

To characterize possible reasons for the reduced LV function under hypoxia conditions, we used immunohistochemistry to determine markers of hypoxia (hypoxia-inducible factor-1α, HIF-1α), nitrosative stress (nitrotyrosine, NT), cellular energy depletion (poly-ADP-ribose polymerase-1, PARP-1) as well as mediators of cell death (poly-ADP-ribose, PAR) and apoptosis (apoptosis-inducing factor, AIF) in the heart. Histological evaluation of the ventricular specimen was carried out according to [53]. To obtain sections including both ventricles and the interventricular septum, the apex of the heart was cut off and fixated in formalin. Then, specimens were embedded in paraffin, and 2 µm sections were cut. The probes were dewaxed and for antigen retrieval treated with 0.3% Triton-X 100 (for nitrotyrosine- and PAR-staining) or cooked in 0.01 M citrate buffer (pH = 6). Afterwards, the histological specimens were blocked with bovine serum albumin (BSA) to saturate unspecific bindings and treated with either mouse monoclonal anti-nitrotyrosine primary antibody (1:100, Merck-Millipore, Darmstadt, Germany), with mouse monoclonal anti-PAR (1:2000), anti-PARP-1 (1:200) or anti-AIF (1:200) primary antibodies (Santa Cruz, Heidelberg, Germany) or with mouse monoclonal anti-HIF-1α primary antibody (1:200, Thermo Scientific, Dreieich, Germany) overnight at 4 °C. Subsequently, specimens were washed in phosphate buffer, and HRP-labeled secondary rabbit anti-mouse antibody (1:200, Sigma-Aldrich, Taufkirchen, Germany) was applied for 2 h. Next, the specimens were washed again, and the peroxidase reaction was carried out using the red chromogen AEC (3-amino-9-ethylcarbazole, Dako, Hamburg, Germany) according to the manufacturer’s instructions. Nuclei of cardiomyocytes were counter-stained with haemalum.

Specimens were embedded in glycerol gelatine (Sigma-Aldrich, Taufkirchen, Germany). The slides were investigated at 400x magnification using a Zeiss Axiolab microscope (Zeiss, Jena, Germany). From each histological slice, 14 photographs were randomly taken and evaluated by a blinded investigator (E.N.). Myocardial regions from the free walls of both the RV and LV were analyzed separately. For measurements in the pictures, the programs ZEN Pro 2011 and ZEN3.3 (both from Zeiss, Jena, Germany) and ImageJ [54] were used. As AIFs, PAR and PARP-1 are located in the nucleus, and the number of nuclei from at least 200 cells per heart was counted. Nuclei with at least 50% red staining were counted as positive nuclei; nuclei with less than 50% red staining were considered to be negative. The ratio of positive nuclei was evaluated as a percentage of the total number of counted nuclei. HIF-1α and NT, which are located in the cytoplasm, were evaluated by determining the red-stained cell area. A threshold value of positive staining was determined using the program ImageJ for each slice. The total cell area exceeding this threshold was measured as positive. After elimination of staining defects and artifacts, the positive area was specified as a percentage of the total area of the free walls of the RV and LV.

### 4.6. Statistical Analysis

Statistical analyses were carried out with the software package SigmaPlot Version 14.0 (Systat Software GmbH, Erkrath, Germany) for Windows. In a first step, we compared the normoxic and hypoxic cohorts. At first, a Shapiro-Wilk test of normality was performed. If data were normally distributed and a Brown–Forsythe test showed equal variances of the cohorts, Student’s *t*-test was applied. In case of unequal variances, we performed Welch’s *t*-test. If the Shapiro-Wilk test of normality failed, a Mann–Whitney rank sum test was used. Then, all groups were statistically compared using analysis of variance (ANOVA) procedures. Again, a Shapiro-Wilk test of normality was performed at first. If it revealed normality, a one-way ANOVA with a post hoc test according to Fisher’s LSD method was applied. If the data were not normally distributed, we used a Kruskal–Wallis ANOVA on ranks with a post hoc test according to Dunn’s method. Both post hoc tests are multiple comparison procedures comparing all possible pairwise mean differences. *p* values < 0.05 were considered significant.

## 5. Conclusions

The results of the present study showed that the hypoxia-induced LV depression continued persistently after 72 h of hypoxia, even though the animals presented with signs of acclimatization to hypoxia such as increased Hb and Hct. The results of the immunohistochemical analyses point towards an energy deficiency of the hypoxic cardiomyocytes, which may account for the depression of LV function. Nitrosative stress and upregulation of PARP-1 as an ATP-consuming DNA repair mechanism are important causes for ATP deficiency. In addition, these processes induce apoptosis, thus further deteriorating the pumping function of the LV. However, nitrosative stress, PARP-1 activation and apoptosis have also been found in the RV, but RV function was not reduced in the hypoxic rats. This indicates that further mechanisms, which are differentially activated in the LV and RV, may also contribute to the hypoxia-induced impairment of cardiac function. Combination of hypoxia and sympathetic blockade further deteriorated LV function but only slightly compared to hypoxic groups with maintained sympathetic efficacy. These results show that hypoxic sympathetic activation at least partially compensates for the compromised cardiac function in hypoxia, thus emphasizing its importance for adaptation to hypoxia. The findings of this study may have important implications for the treatment of patients with multimorbidity.

## Figures and Tables

**Figure 1 ijms-24-11417-f001:**
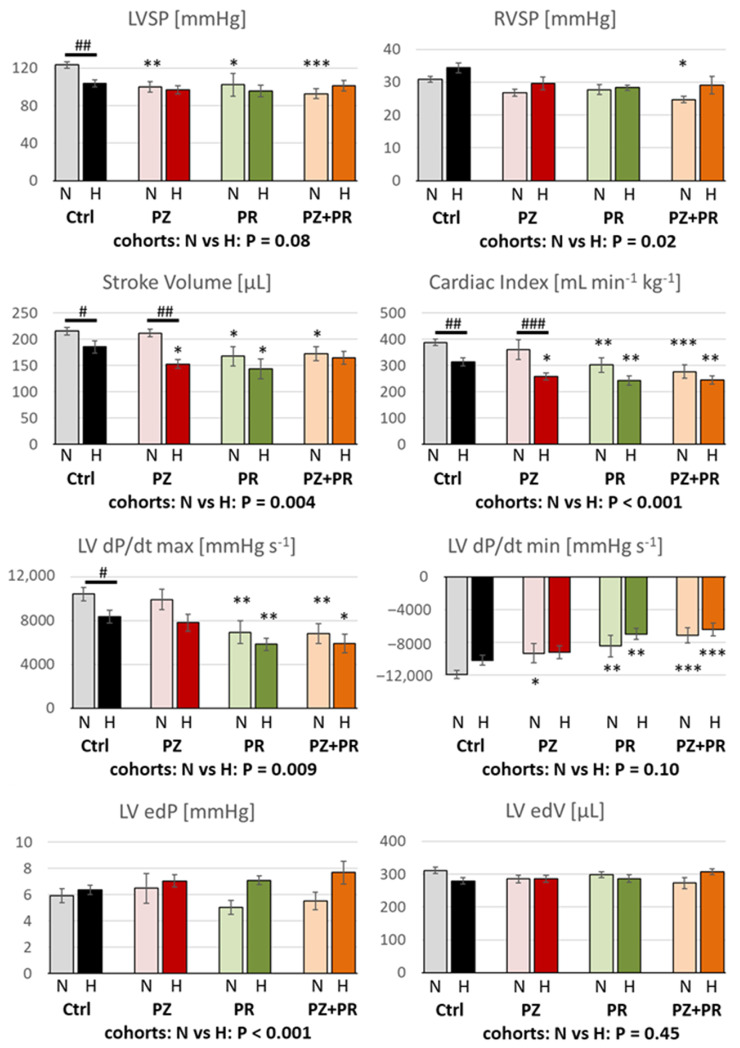
LVSP = left ventricular systolic pressure (*n =* 8–15); RVSP = right ventricular systolic pressure (*n =* 8–16); LV dP/dt max, LV dP/dt min = left ventricular maximal velocity of contraction and relaxation, respectively (*n =* 7–15); LV edP = left ventricular end-diastolic pressure (*n =* 6–15); LV edV = left ventricular end-diastolic volume (*n =* 6–15). For group abbreviations, please see Section 4. Data are given as means ± SEM. Significance marks: significant differences between corresponding normoxic and hypoxic groups: # *p* < 0.05; ## *p* < 0.01; ### *p* < 0.001; significant differences between blocker groups and the related (normoxic or hypoxic) control are marked with asterisks: * *p* < 0.05; ** *p* < 0.01; *** *p* < 0.001.

**Figure 2 ijms-24-11417-f002:**
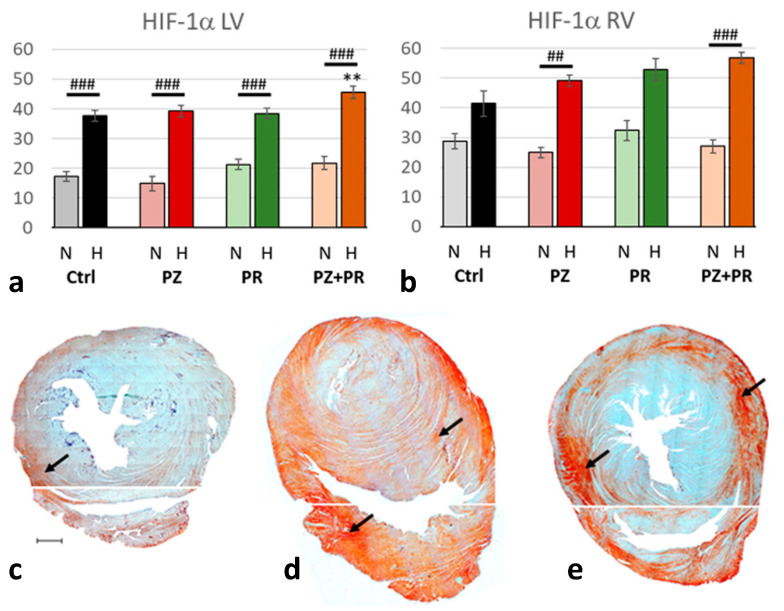
(**a**,**b**): Abundance of hypoxia-inducible factor (HIF)-1α in the LV (**a**) and RV (**b**) expressed as percentage of positive area related to the total area of the free walls of LV and RV, respectively. Data are presented as means ± SEM; *n =* 8 for each group. For group abbreviations, please see Section 4. Significant difference between normoxic and hypoxic cohorts: *p* < 0.001. Significant differences between corresponding normoxic and hypoxic groups: ## *p* < 0.01; ### *p* < 0.001; significant differences between blocker groups and the related (normoxic or hypoxic) control are marked with asterisks: ** *p* < 0.01. (**c**–**e**): Representative immunohistochemical images from N-Ctrl (**c**), H-Ctrl (**d**) and H-PZ+PR (**e**) hearts. All images are shown in the same magnification with the scale bar in part c indicating 1000 µm. Red staining indicates HIF-1α-positive cells (examples are marked by arrows). Separate photographs have been taken from LV and RV.

**Figure 3 ijms-24-11417-f003:**
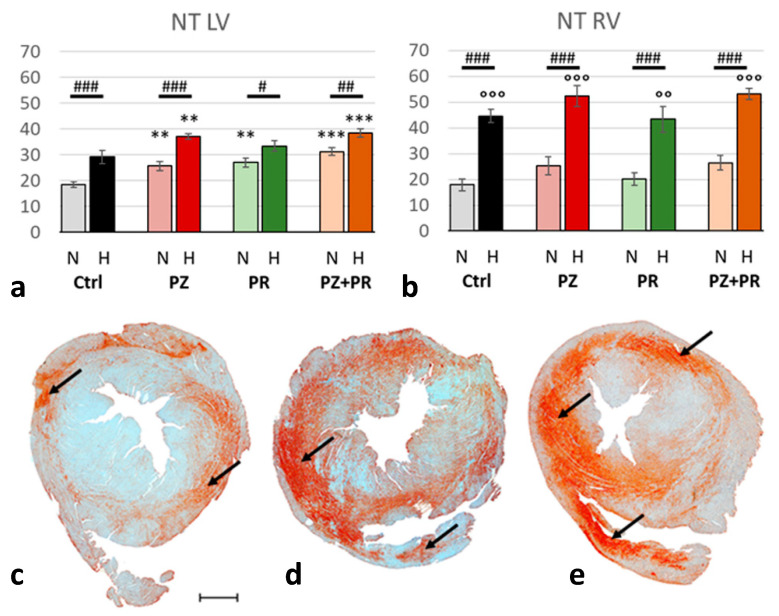
(**a**,**b**): Abundance of nitrotyrosine (NT) in the LV (**a**) and RV (**b**) expressed as percentage of positive area related to the total area of the free walls of LV and RV, respectively. Data are presented as means ± SEM; *n =* 8 for each group. For group abbreviations, please see Section 4. Significant difference between normoxic and hypoxic cohorts: *p* < 0.001. Significant differences between corresponding normoxic and hypoxic groups: # *p* < 0.05; ## *p* < 0.01; ### *p* < 0.001; significant differences between blocker groups and the related (normoxic or hypoxic) control are marked with asterisks: ** *p* < 0.01; *** *p* < 0.001; significant differences of RV to LV are marked with circles: °° *p* < 0.01; °°° *p* < 0.001. (**c**–**e**): Representative immunohistochemical images from N-Ctrl (**c**), H-Ctrl (**d**) and H-PZ+PR (**e**) hearts. All images are shown in the same magnification with the scale bar in part c indicating 1000 µm. Red staining indicates NT-positive cells (examples are marked by arrows).

**Figure 4 ijms-24-11417-f004:**
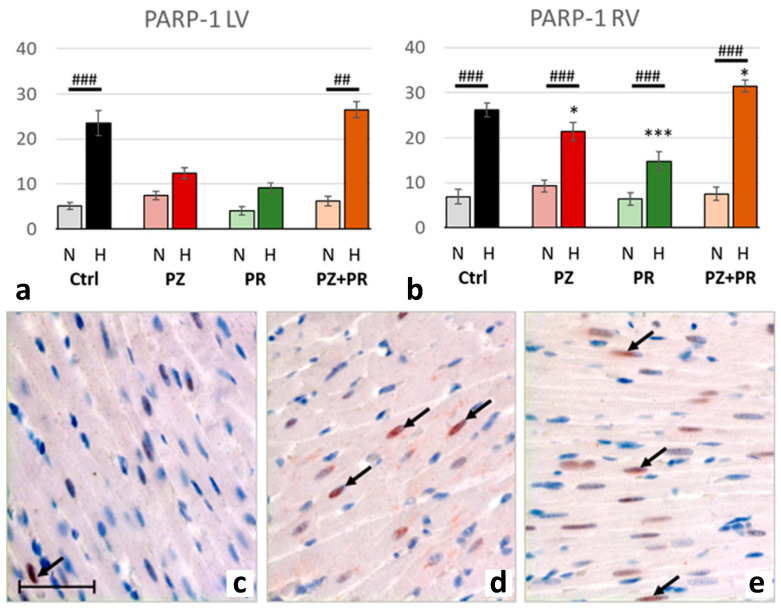
(**a**,**b**): Abundance of poly-ADP-ribose-polymerase (PARP) 1 in the LV (**a**) and RV (**b**) expressed as percentage of positive cell nuclei related to the total number of counted nuclei in the LV and RV, respectively. Data are presented as means ± SEM; *n =* 8 for each group. For group abbreviations, please see Section 4. Significant difference between normoxic and hypoxic cohorts: *p* < 0.001. Significant differences between corresponding normoxic and hypoxic groups: ## *p* < 0.01; ### *p* < 0.001; significant differences between blocker groups and the related (normoxic or hypoxic) control are marked with asterisks: * *p* < 0.05; *** *p* < 0.001. (**c**–**e**): Representative immunohistochemical images from N-Ctrl (**c**), H-Ctrl (**d**) and H-PZ+PR (**e**) hearts. All images are shown in the same magnification with the scale bar in part c indicating 100 µm. PARP-1-positive nuclei are stained in red (examples are marked by arrows).

**Figure 5 ijms-24-11417-f005:**
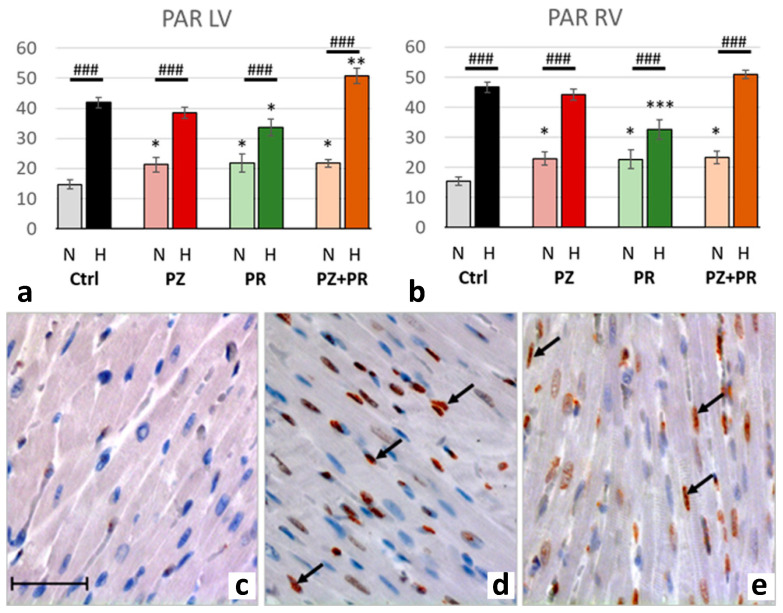
(**a**,**b**): Abundance of poly-ADP-ribose (PAR) in the LV (**a**) and RV (**b**) expressed as percentage of positive cell nuclei related to the total number of counted nuclei in the LV and RV, respectively. Data are presented as means ± SEM; *n =* 8 for each group. For group abbreviations, please see Section 4. Significant difference between normoxic and hypoxic cohorts: *p* < 0.001. Significant differences between corresponding normoxic and hypoxic groups: ### *p* < 0.001; significant differences between blocker groups and the related (normoxic or hypoxic) control are marked with asterisks: * *p* < 0.05; ** *p* < 0.01; *** *p* < 0.001. (**c**–**e**): Representative immunohistochemical images from N-Ctrl (**c**), H-Ctrl (**d**) and H-PZ+PR (**e**) hearts. All images are shown in the same magnification with the scale bar in part c indicating 100 µm. PAR-positive nuclei are stained in red (examples are marked by arrows).

**Figure 6 ijms-24-11417-f006:**
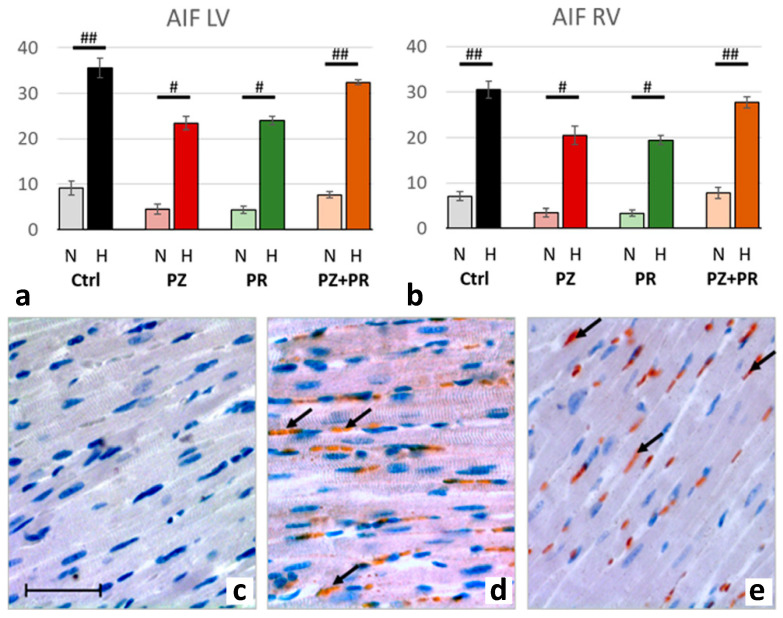
(**a**,**b**): Abundance of apoptosis-inducing factor (AIF) in the LV (**a**) and RV (**b**) expressed as percentage of positive cell nuclei related to the total number of counted nuclei in the LV and RV, respectively. Data are presented as means ± SEM; *n =* 8 for each group. For group abbreviations, please see Section 4. Significant difference between normoxic and hypoxic cohorts: *p* < 0.001. Significant differences between corresponding normoxic and hypoxic groups: # *p* < 0.05; ## *p* < 0.01. (**c**–**e**): Representative immunohistochemical images from N-Ctrl (**c**), H-Ctrl (**d**) and H-PZ+PR (**e**) hearts. All images are shown in the same magnification with the bar in part c indicating 100 µm. AIF-positive nuclei are stained in red (examples are marked by arrows).

**Table 1 ijms-24-11417-t001:** Nutritional condition.

Cohort	Normoxic Cohort:				Hypoxic Cohort:			
Group (*n*)	N-Ctrl (7–10)	N-PZ (6–8)	N-PR (7–8)	N-PZ+PR (8)	H-Ctrl (8–10)	H-PZ (8–14)	H-PR (13–14)	H-PZ+PR (9–10)
BW change (% baseline)	**−2.3 (−5.0; 0.5)**				**−12.7 (−14.7; −9.9) ###**			
	−1.7 (−3.7; 4.4)	−4.2 (−7.7; −2.1)	−1.7 (−3.7; 4.4)	−4.2 (−7.7; −2.1)	−1.7 (−3.7; 4.4)	−4.2 (−7.7; −2.1)	−1.7 (−3.7; 4.4)	−4.2 (−7.7; −2.1)
Total food uptake (g)	**34.8 (23.4; 43.2)**				**14.2 (9.2; 19.4) ###**			
	43.2 (38.6; 47.4)	24.6 (20.4; 37.6)	43.2 (38.6; 47.4)	24.6 (20.4; 37.6)	43.2 (38.6; 47.4)	24.6 (20.4; 37.6)	43.2 (38.6; 47.4)	24.6 (20.4; 37.6)
c (Glucose) (mmol/L)	**8.6 (8.4; 9.8)**				**12.4 (10.0; 14.2) ###**			
	8.4 (7.4; 8.6)	9.0 (8.4; 12.0)	8.4 (7.4; 8.6)	9.0 (8.4; 12.0)	8.4 (7.4; 8.6)	9.0 (8.4; 12.0)	8.4 (7.4; 8.6)	9.0 (8.4; 12.0)

Data are given as medians (25th; 75th percentiles) (*n* = number of values per group). Cohort medians (25th; 75th percentiles) are presented in bold characters. Cohort/group labels: N = normoxia; H = hypoxia; Ctrl = control group (0.9% saline infusion); PZ = prazosin; PR = propranolol. More detailed information on animal groups and drug doses is given in Section 4. BW change = change in body weight (BW2 − BW1); c (Glucose) = blood glucose concentration. Significant differences from the hypoxic cohort or groups to the normoxic cohort or to the corresponding normoxic group: ### *p* < 0.001.

**Table 2 ijms-24-11417-t002:** Oxymetry and blood gases.

Cohort	Normoxic Cohort:				Hypoxic Cohort:			
Group (*n*)	N-Ctrl (6–9)	N-PZ (6–8)	N-PR (6–8)	N-PZ+PR (6–8)	H-Ctrl (6–10)	H-PZ (6–10)	H-PR (7–14)	H-PZ+PR (6–9)
c (Hb) (g/mL)	**13.5 ± 0.2**				**16.2 ± 0.1 ###**			
	14.1 ± 0.2	13.1 ± 0.6	13.5 ± 0.4	13.1 ± 0.3	15.9 ± 0.2 ###	16.3 ± 0.3 ###	15.9 ± 0.3 ###	16.8 ± 0.3 ###
Hematocrit (%)	**41.5 (39.6; 43.4)**				**49.2 (47.6; 50.7) ###**			
	42.3 (41.2; 44.6)	39.9 (38.0; 40.4)	42.3 (41.2; 44.6)	39.9 (38.0; 40.4)	42.3 (41.2; 44.6)	39.9 (38.0; 40.4)	42.3 (41.2; 44.6)	39.9 (38.0; 40.4)
S_a_O_2_ (%)	**91.8 ± 0.9**				**84.5 ± 1.4 ###**			
	92.8 ± 0.9	92.4 ± 2.4	90.2 ± 3.7	91.1 ± 1.3	82.1 ± 2.1 ##	80.8 ± 2.8 ##	89.8 ± 1.9 *	83.4 ± 3.3 #
Arterial pH	**7.44 (7.42; 7.50)**				**7.33 (7.29; 7.38) ###**			
	7.44 (7.43; 7.49)	7.47 (7.40; 7.53)	7.44 (7.43; 7.49)	7.47 (7.40; 7.53)	7.44 (7.43; 7.49)	7.47 (7.40; 7.53)	7.44 (7.43; 7.49)	7.47 (7.40; 7.53)
Arterial pCO_2_ (mmHg)	**38.0 ± 1.1**				**30.1 ± 1.1 ###**			
	37.4 ± 1.8	40.2 ± 2.7	33.4 ± 2.7	39.9 ± 1.7	32.6 ± 2.9	31.1 ± 2.0 #	30.2 ± 1.4	25.0 ± 1.4 ### *
Base excess	**+2.8 ± 0.3**				**−10.4 ± 0.8 ###**			
	+2.4 ± 0.6	+2.9 ± 0.7	+2.6 ± 0.1	+5.1 ± 0.0	−16.3 ± 1.9 ###	−10.7 ± 0.0 ### *	−9.6 ± 0.8 ### ***	−9.0 ± 0.9 ### ***

Normally distributed data are given as means ± SEM; if data were not normally distributed, the medians (25th; 75th percentiles) are presented (*n* = number of values per group). Cohort means (± SEM) and medians (25th; 75th percentiles) are presented in bold characters. Cohort/group labels: N = normoxia; H = hypoxia; Ctrl = control group (0.9% saline infusion); PZ = prazosin; PR = propranolol. c (Hb) = Hemoglobin concentration; S_a_O_2_ = arterial oxygen saturation; pCO_2_ = partial pressure of carbon dioxide. Significant differences from the hypoxic cohort or groups to the normoxic cohort or to the corresponding normoxic group: # *p* < 0.05; ## *p* < 0.01; ### *p* < 0.001; significant differences between blocker groups and the related control are marked with asterisks: * *p* < 0.05; *** *p* < 0.001.

**Table 3 ijms-24-11417-t003:** Hemodynamic results.

Cohort	Normoxic Cohort:				Hypoxic Cohort:			
Group (*n*)	N-Ctrl (9–14)	N-PZ (6–7)	N-PR (6–8)	N-PZ + PR (6–8)	H-Ctrl (14–16)	H-PZ (9–13)	H-PR (7–12)	H-PZ + PR (9)
EF [%]	**56.9 ± 1.8**				**48.8 ± 1.7** ##			
	61.8 ± 1.2	62.6 ± 1.5	49.5 ± 5.6 *	50.7 ± 4.3 *	54.1 ± 2.9 #	45.9 ± 2.5 ## *	42.7 ± 4.2 *	48.4 ± 3.4
SW [mmHg µL]	**15,001 ± 1146**				**10,954 ± 664** ##			
	18512 ± 2174	16857 ± 1104	12476 ± 1560 *	10467 ± 1532 ***	12852 ± 1513 ##	9577 ± 939 ##	9151 ± 1164	11830 ± 1095
HR [min^−1^]	**426.0 ± 7.7**				**390.0 ± 6.2** ###			
	441.8 ± 8.1	458.1 ± 11.2	423.9 ± 9.3	374.5 ± 21.1 ***	410.4 ± 8.3 #	407.4 ± 11.1 ##	359.6 ± 11.9 ### ***	367.7 ± 13.9 **
LV esE [mmHg/µL]	**0.33 (0.26; 0.50)**				**0.36 (0.24; 0.71)**			
	0.36 (0.24; 0.53)	0.28 (0.26; 0.34)	0.44 (0.37; 0.55)	0.24 (0.17; 0.50)	0.34 (0.26; 0.66)	0.57 (0.28; 0.76)	0.33 (0.25; 0.58)	0.25 (0.18; 0.72)
RV dP/dtmax [mmHg/s]	**2053 (1602; 2488)**				**1919 (1530; 2687)**			
	2154 (2072; 2448)	2441 (2001; 3233)	1888 (1609; 3348)	1473 (1390; 1719)	2511 (1823; 3118)	2238 (1799; 3049)	1568 (1503; 1919)	1402 (1158; 1631) *
RV dP/dtmin [mmHg/s]	**−1711 (−2145; −1215)**				**−1618 (−2138; −1274)**			
	−1896 (−2501; −1727)	−1863 (−2164; −1166)	−1409 (−1774; −1228)	−1208 (−1454; −1053)	−2088 (−2384; −1487)	−1856 (−2291; −1236)	−1379 (−1681; −1077)	−1333 (−1778; −1098)
MAP [mmHg]	**96.4 ± 3.5**				**91.0 ± 2.4**			
	109.6 ± 3.5	91.0 ± 5.7	92.7 ± 11.6	84.7 ± 5.0	91.4 ± 3.6	91.2 ± 4.8	87.1 ± 5.9	95.2 ± 5.7
TPR [mmHg min kg s^−1^]	**0.28 (0.25; 0.33)**				**0.31 (0.29; 0.41)** #			
	0.28 (0.25; 0.32)	0.25 (0.24; 0.28)	0.28 (0.23; 0.46)	0.31 (0.26; 0.38)	0.30 (0.25; 0.33)	0.30 (0.27; 0.46)	0.33 (0.30; 0.43)	0.40 (0.30; 0.51)

Normally distributed data are given as means ± SEM; if data were not normally distributed, the medians (25th; 75th percentiles) are presented (*n* = number of values per group). Cohort means (± SEM) and medians (25th; 75th percentiles) are presented in bold characters. Cohort/group labels: N = normoxia; H = hypoxia; Ctrl = control group (0.9% saline infusion); PZ = prazosin; PR = propranolol. LV, RV = left/right ventricle; EF = ejection fraction; esE = end-systolic elastance; SW = stroke work; dP/dt max = maximal velocity of contraction; dP/dt min = maximal velocity of relaxation; HR = heart rate; MAP = mean aortic pressure; TPR = total peripheral resistance. Significant differences from the hypoxic cohort or groups to the normoxic cohort or to the corresponding normoxic group: # *p* < 0.05; ## *p* < 0.01; ### *p* < 0.001; significant differences between blocker groups and the related control (N-Ctrl or H-Ctrl) are marked with asterisks: * *p* < 0.05; ** *p* < 0.01; *** *p* < 0.001.

**Table 4 ijms-24-11417-t004:** Animal groups.

Group	Number of Animals
N-Ctrl	14
H-Ctrl	18
N-PZ	8
H-PZ	14
N-PR	8
H-PR	14
N-PZ+PR	8
H-PZ+PR	10

Group labels: N = normoxia; H = hypoxia; Ctrl = control group (0.9% saline infusion); PZ = prazosin infusion, 0.1 mg kg^−1^ h^−1^; PR = propranolol infusion, 0.16 mg kg^−1^ h^−1^; PZ+PR = infusion with prazosin + propranolol, same doses as described above.

## Data Availability

Data are available on request from the corresponding author.

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
