# Peer review of "Effects of Normobaric Hypoxia and Adrenergic Blockade over 72 h on Cardiac Function in Rats"

_ijms, 2023, doi:10.3390/ijms241411417_

Round 1

Reviewer 1 Report

Comments:

  1. The study provides valuable insights into the effects of prolonged hypoxia on left ventricular (LV) function in rats. However, the manuscript would benefit from a clearer presentation of the research objectives and hypotheses. It is important to explicitly state the research questions being addressed and the specific aims of the study in the introduction.
  2. The sample size of the study (94 rats) appears to be adequate, but it would be helpful to provide additional details regarding the randomization and allocation of animals to the different experimental groups in methods. This information would enhance the transparency and reproducibility of the study.
  3. While the manuscript mentions the investigation of biomarkers of nitrosative stress and apoptosis as potential causes of hypoxic LV depression, it does not provide sufficient details on the methods used for these analyses. Additional information on the specific biomarkers assessed, the techniques employed, and the rationale behind their selection would strengthen the research findings.
  4. The manuscript briefly mentions the effects of adrenergic blockers on cardiac function under normoxic and hypoxic conditions. It would be beneficial to provide more detailed information on the specific adrenergic blockers used, their dosages, and the rationale for their selection. This information would help readers better understand the experimental design and interpretation of the results.
  5. The findings regarding the recovery of LV function after 11-24 hours of hypoxic exposure and the slight recovery observed after 72 hours are intriguing. However, it is important to discuss the potential limitations of these findings. For example, did the duration of recovery vary among individual animals, and if so, what factors could have contributed to this variability?
  6. The manuscript  mentions the immunohistochemical analyses revealing nitrosative stress, ATP deficiency, and induction of apoptosis as potential mechanisms underlying hypoxic LV depression. However, the specific methods and techniques used for these analyses need to be detailed. Providing details on the antibodies used, staining protocols, and quantification methods would enhance the scientific rigor and reproducibility of the study.
  7. It is crucial to discuss the clinical relevance of the findings in the context of human cardiac function and hypoxia-related conditions. How do the observed effects in rats align with the pathophysiology of hypoxia-induced LV dysfunction in humans? Including a brief discussion on the translational implications would strengthen the significance and impact of the study.
  8. The manuscript would benefit from a clear and concise summary of the key findings and their implications in a bit more detail in the conclusion section. This would help readers easily grasp the main takeaways of the study without having to revisit the entire manuscript.
  9. Lastly, there are a few minor grammatical and formatting errors throughout the manuscript. It is recommended to thoroughly proofread the manuscript for clarity and correctness before finalizing it for publication.

Overall, the study presents important findings on the effects of prolonged hypoxia on LV function and provides insights into potential underlying mechanisms. Addressing the above-mentioned points would enhance the clarity, scientific rigor, and impact of the manuscript.

There are a few minor grammatical and formatting errors throughout the manuscript. It is recommended to thoroughly proofread the manuscript for clarity and correctness before finalizing it for publication

Author Response

Responses to Reviewer #1:

We thank this reviewer for his/her valuable and constructive comments on our manuscript, which helped to improve the quality of the text.

  1. The study provides valuable insights into the effects of prolonged hypoxia on left ventricular (LV) function in rats. However, the manuscript would benefit from a clearer presentation of the research objectives and hypotheses. It is important to explicitly state the research questions being addressed and the specific aims of the study in the introduction.

Response: We have improved the text and stated our research questions and specific aims more clearly. We have changed the last paragraph of the Introduction section as follows:

“This study was designed to examine the effects of prolonged exposure to normobaric hypoxia over 72 h on rats. The main question was whether adaptation to hypoxia is detectable after 72 h and may be associated with recovery of cardiac function. The second question was directed to the possible reasons of hypoxic LV depression. Specifically, we hypothesized that nitrosative stress, ATP deficiency and apoptosis might be important reasons for the hypoxia-induced reduction in LV pump function. Finally, we were interested in the role of hypoxic sympathetic activation. For this reason, we studied whether adrenergic blockade would further deteriorate the general state of the animals and their cardiac function.

Markers of the nutritional condition, metabolism and blood gases were determined to assess the general state of the animals. Hemoglobin (Hb) concentration, hematocrit and the expression of HIF-1a served as markers of adaptation. LV and RV catheterization provided parameters of cardiac function. We determined the expression of nitrotyrosine, PARP-1, PAR and AIF as markers of nitrosative stress, ATP deficiency and apoptosis, respectively.”

  1. The sample size of the study (94 rats) appears to be adequate, but it would be helpful to provide additional details regarding the randomization and allocation of animals to the different experimental groups in methods. This information would enhance the transparency and reproducibility of the study.

Response: We have added some text into the Methods section (at the end of subsection 4.2.) explaining study design and allocation of animals to the experimental groups in more detail. In brief, the original study design only comprised 5 groups. We expanded the groups after receiving the possibility to perform blood gas analyses (this is explained in more detail in a new Limitations subsection). Allocation of animals to the cohorts and to the groups was done by random numbers.

  1. While the manuscript mentions the investigation of biomarkers of nitrosative stress and apoptosis as potential causes of hypoxic LV depression, it does not provide sufficient details on the methods used for these analyses. Additional information on the specific biomarkers assessed, the techniques employed, and the rationale behind their selection would strengthen the research findings.

Response: We have used immunohistochemistry to investigate the markers in the heart that might elucidate possible reasons for the LV dysfunction under hypoxia. We have added some more information on the meaning of these markers in the Methods section (subsection 4.5. “Immunohistochemistry”). Generation of the histological slices including preparation of the histological specimens, treatment with the specific antibodies as well as staining of the slices has been described in this subsection.  Moreover, we have inserted a short subsection 4.4. “Sampling of materials”. We think, however, that the detailed staining protocols would exceed the scope of this paper, that is why we have provided just a truncated description of the preparation and staining procedures. The analysis of the slices has also been described. We have added some more details to make the detection of positive nuclei or positive cell area more reproducible for the reader.

  1. The manuscript briefly mentions the effects of adrenergic blockers on cardiac function under normoxic and hypoxic conditions. It would be beneficial to provide more detailed information on the specific adrenergic blockers used, their dosages, and the rationale for their selection. This information would help readers better understand the experimental design and interpretation of the results.

Response: We have used the a-adrenergic blocker prazosin (PZ) at a dose of 0.1 mg kg-1 h-1); further, the b-adrenergic blocker propranolol (PR) at a dose of 0.16 mg kg-1 h-1; finally, a combination of the two blockers (PZ+PR, 0.1 + 0.16 mg kg-1 h-1, respectively). This has been described in the Methods section (subsection 4.2. Study protocol). We have chosen these drugs as they are still in clinical use in patients and may serve as a model for patients in a hypoxic condition who are treated with adrenergic blockers. We have stated this rationale in subsection 4.2.:

“These drugs are in clinical use in patients, this means that the use of these drugs may serve as a model for patients in a hypoxic condition, e.g., due to pulmonary diseases, under treatment with adrenergic blockers. For instance, b-blockers are administered to patients with coronary heart disease or after myocardial infarction. Among the b-blockers, we have chosen propranolol as it blocks both b1- and b2-adrenoceptors, thus inducing both cardiac and vascular effects. Prazosin is a blocker of a1-adrenoceptors that is applied in the treatment of arterial hypertension and of arterial ischemic conditions. Sympathetic vasoconstrictor effects are mainly mediated via stimulation of a1-adrenoceptors.”

  1. The findings regarding the recovery of LV function after 11-24 hours of hypoxic exposure and the slight recovery observed after 72 hours are intriguing. However, it is important to discuss the potential limitations of these findings. For example, did the duration of recovery vary among individual animals, and if so, what factors could have contributed to this variability?

Response: In the previous study (Bölter et al., 2019), we observed a decrease in LVSP and LV dP/dt max by almost 25% and 50% of normoxic values, respectively, after 6 h of hypoxia, as has been stated in the Discussion section of the present manuscript. After 16 h of hypoxia, these values were still at the same level (even marginally lower), but there was a mild re-increase by 24 h. This means, that LV inotropic function has slightly improved between the 16th and 24th hour of exposure. However, there were overlaps between the two groups, and the difference between the 16-h- and 24-h-hypoxic groups was not significant (in the direct comparison). In the 6-h- and 16-h-hypoxic groups, the values were significantly lower than in the respective normoxic groups and/or in the 1.5-h-hypoxic group, while there was no significant difference in the 24-h-hypoxic group. Therefore, we wrote in that paper: “A tendency of recovery occurred after 24 h of hypoxia”. Similarly, we stated in the present manuscript: “…but showed a slight tendency towards recovery after 24 h of hypoxia.” (Introduction section, first paragraph). We would speculate that processes of acclimatization to hypoxia might have started as early as after about one day of hypoxia. That is why one of the main questions of the present study was whether a clear recovery of LV function can be observed after 72 h of hypoxia. We have stated this more clearly now in the last paragraph of the Introduction section. Such a recovery has not been confirmed in the present study. After 72 h, LV function was in the same range as after 24 h (as found in the previous study by Bölter et al., 2019).  However, the lower number of premature deaths of animals during the final thiopental narcosis indicated a stabilization of the animals’ condition after 72 h of hypoxia. This has been inserted into the Discussion section. We assume that the improvement of the animals’ condition is probably due to acclimatization to hypoxia (as reflected e.g., by elevated Hb and Hct) even though LV function had not improved, which has also been inserted into the Discussion section.  The results of the immunohistochemical analyses provide some possible explanations (nitrosative stress, energy deficiency, apoptosis) for the absence of a recovery of the LV function despite clear signs of general acclimatization to hypoxia (increased Hct and Hb). This has been inserted into the Discussion section (last paragraph of the subsection 3.2).

  1. The manuscript mentions the immunohistochemical analyses revealing nitrosative stress, ATP deficiency, and induction of apoptosis as potential mechanisms underlying hypoxic LV depression. However, the specific methods and techniques used for these analyses need to be detailed. Providing details on the antibodies used, staining protocols, and quantification methods would enhance the scientific rigor and reproducibility of the study.

Response: This information is given in subsection 4.4. “Immunohistochemistry” in the Methods section, and we have added some more details (please see our response to your point 3).

  1. It is crucial to discuss the clinical relevance of the findings in the context of human cardiac function and hypoxia-related conditions. How do the observed effects in rats align with the pathophysiology of hypoxia-induced LV dysfunction in humans? Including a brief discussion on the translational implications would strengthen the significance and impact of the study.

Response: This may be a relevant problem in many elderly people with multimorbidity such as cardiovascular diseases and COPD. In many cases, treatment with adrenergic agonists or adrenergic blockers may be indicated for several reasons. However, a hypoxic/hypoxemic condition may critically aggravate potential adverse effects of some medication such as beta-blockers on a pre-damaged heart and, moreover, on the whole organism.

We have inserted some text referring to clinical application of adrenergic blockers into the methods section (subsection 2.2 “Protocol”) to explain our rationale for choosing the drugs.

In addition, we have extended the last paragraph of the Discussion subsection 3.2. to emphasize the clinical relevance of our results as follows:

“… the combination of hypoxia and adrenergic blockade may create a vulnerable condition for the organism. If combined with additional stressors or diseases, this vulnerable condition may easily result in fatal consequences. This illustrates the importance of hypoxic sympathetic activation for adaptation to hypoxia. If sympathetic activity or cardiac sympathetic innervation is reduced or even abolished in a hypoxic or hypoxemic condition, the risk for cardiac decompensation is high. This may have important implications for patients with multimorbidity, e.g., patients with both pulmonary and cardiovascular diseases, and especially for their treatment, e.g., with b-blockers. Our results show, in contrast, that hypoxic sympathetic activation can at least partially compensate for the compromised cardiac function under hypoxia.”

  1. The manuscript would benefit from a clear and concise summary of the key findings and their implications in a bit more detail in the conclusion section. This would help readers easily grasp the main takeaways of the study without having to revisit the entire manuscript.

Response: We have refined the Conclusions section and stated the main results and their implications in a more clear and pronounced manner.

  1. Lastly, there are a few minor grammatical and formatting errors throughout the manuscript. It is recommended to thoroughly proofread the manuscript for clarity and correctness before finalizing it for publication.

 Response: We have carefully proofread and corrected the manuscript. Moreover, an English language specialist checked and corrected the linguistic errors in the manuscript.

Overall, the study presents important findings on the effects of prolonged hypoxia on LV function and provides insights into potential underlying mechanisms. Addressing the above-mentioned points would enhance the clarity, scientific rigor, and impact of the manuscript.

Comments on the Quality of English Language

There are a few minor grammatical and formatting errors throughout the manuscript. It is recommended to thoroughly proofread the manuscript for clarity and correctness before finalizing it for publication

Reviewer 2 Report

Neubert E. and colleagues reported the effects of normobaric hypoxia and adrenergic blockade over 72 h on cardiac function in rats. This work is interesting, but some questions and suggestions should addressed before its publication.

-In the abstract section, the authors should add a phrase as introduction about the relationship between hypoxia and adrenergic blockade on cardiac function. A general conclusion related to the results obtained is also needed. In addition, in this section and throughout the manuscript, there are several very long sentences, thus some commas or semicolon should be used to make the phrases clearer.

-There are some grammatical errors and typos in the manuscript.

-Please define acronyms where they are used for the first time (e.g., LV, HIF-1, and RV).

-Why were the groups of rats included with different n?

-Please indicate if the data analyzed with Mann-Whitney were expressed as median ± quartile (indicate in graph or figure).

-The authors should mention figures and tables during the description of the results, including the values with which they are compared. Moreover, there is no table 4.

-In the first tables, it is difficult to understand the statistical because symbols used are not clear. Please indicate the groups being compared. In addition, authors should add the n for each rat group analyzed. Why are all mice compared as normoxic cohort vs. hypoxic cohort, since in each cohort there are different rat groups with different experimental conditions?

-The Kaplan Meier Curve should used to estimate the survival function for each cohort of rats.

-Clarify if in figure 1 and 4 the symbols represent p<0.05. In figure 1, the appropriate units should also be indicated on each graph (e.g., LVSP in mmHg, etc).

-In figures 2, 3, 4, 5, and 6, panels with immunohistochemical images, authors should indicate with arrows the positive signals, respectively.

-In the next phrase (page 11, line 335-336): “HIF-1 is a heterodimeric protein consisting of HIF-1α and HIF-1 α subunits,” Please correct this information because one is α and the other is β.

-The conclusion should be after the discussion section, please change the order.

-The authors should add the limitations of the study.

In general, the language english is fine, but there are some grammatical errors and typos in the manuscript. 

Author Response

Responses to Reviewer #2:

We thank this reviewer for his/her valuable and constructive comments on our manuscript, which helped to improve the quality of the text.

Neubert E. and colleagues reported the effects of normobaric hypoxia and adrenergic blockade over 72 h on cardiac function in rats. This work is interesting, but some questions and suggestions should addressed before its publication.

  1. In the abstract section, the authors should add a phrase as introduction about the relationship between hypoxia and adrenergic blockade on cardiac function. A general conclusion related to the results obtained is also needed. In addition, in this section and throughout the manuscript, there are several very long sentences, thus some commas or semicolon should be used to make the phrases clearer.

Response: We have changed the Abstract according to this reviewer’s proposals. Moreover, we have refined very long sentences by shortening them or inserting commas.

  1. There are some grammatical errors and typos in the manuscript.

Response: We have carefully proofread and corrected the manuscript. Moreover, an English language specialist checked and corrected the linguistic errors in the manuscript.

  1. Please define acronyms where they are used for the first time (e.g., LV, HIF-1, and RV).

Response: We have checked the text and defined all abbreviations at their first appearance.

  1. Why were the groups of rats included with different n?

Response: We have extended the description of the study protocol. In particular, we have explained the different n in the rat groups. We have inserted additional text into the Methods section as follows:

“The minimum number of animals within each group was defined to be n = 8. … Initially, the design of the study only comprised 5 groups (i.e., 40 animals): a normoxic and a hypoxic control group as well as 3 hypoxic groups with adrenergic blockers. … After the first part of these experiments was done, we decided to add 3 groups with normoxic animals receiving adrenergic blockers to allow a direct comparison between normoxic and hypoxic animals with the same blocker treatment. … Due to complications in the experimental course and measurement failures (see Limitations section), we often obtained incomplete datasets. Therefore, we increased the number of animals per group until there were at least 6-8 values per group for each measurement. In addition, we included 4 normoxic and 8 hypoxic control animals from the pre-tests into the final evaluation.”

  1. Please indicate if the data analyzed with Mann-Whitney were expressed as median ± quartile (indicate in graph or figure).

Response: For reasons of homogeneity, all data presented in the figures and tables are given as means ± SEM. This has been stated now in all legends to the figures and tables.

  1. The authors should mention figures and tables during the description of the results, including the values with which they are compared. Moreover, there is no table 4.

Response: We have inserted the references to the respective Figures and Tables and the groups/values with which we have compared our results into the text. Table 4 was incorrectly labelled; we apologize for this mistake; it has been corrected now.

  1. In the first tables, it is difficult to understand the statistical because symbols used are not clear. Please indicate the groups being compared. In addition, authors should add the n for each rat group analyzed. Why are all mice compared as normoxic cohort vs. hypoxic cohort, since in each cohort there are different rat groups with different experimental conditions?

Response: We have revised and standardized the significance symbols so that they are uniform throughout the manuscript, and we have indicated the groups being compared.  Moreover, we have added the n into the Tables and Figure legends.

We observed a clear effect of hypoxia on most markers of general state, blood gases, metabolism, and cardiac function. We wished to highlight this important result; therefore, we compared the normoxic and hypoxic cohorts. In addition, we compared all 8 groups with each other using a multiple comparison procedure (ANOVA and post-hoc test) as described in the Methods section.

  1. The Kaplan Meier Curve should used to estimate the survival function for each cohort of rats.

Response: We think that Kaplan Meier curves are not appropriate to characterize the survival time of the animals. The duration of the experiment was pre-defined to be 72 h, and all animals survived at least until administration of the final narcosis (i.e., at least until 45 min before the pre-defined end of the experiment). Even more, we would speculate that all animals would have survived without thiopental narcosis and heart catheterization. We have explained this in some more detail in the Results section in a new separate subsection (2.2. Complications prior to or during hemodynamic measurements) emphasizing the fact that those animals died under thiopental narcosis and during hemodynamic measurements.

Moreover, we have emphasized the importance of narcosis and interventional hemodynamic examination for premature deaths in the Discussion section:

“We assumed that additional stress due to narcosis and heart catheterization finally resulted in decompensation and acute RV failure [1]. The lower incidence of premature deaths in the present study (11% of all hypoxic animals) indicates that the situation of the animals had stabilized after 3 days of hypoxia.”

  1. Clarify if in figure 1 and 4 the symbols represent p<0.05. In figure 1, the appropriate units should also be indicated on each graph (e.g., LVSP in mmHg, etc).

Response: We have inserted significance symbols indicating the range of p values (p<0.05, p<0.01, p<0.001) into all graphs. In Figure 1, we have inserted the units into the title of each panel.

  1. In figures 2, 3, 4, 5, and 6, panels with immunohistochemical images, authors should indicate with arrows the positive signals, respectively.

Response: We have inserted arrows indicating the positive signals.

  1. In the next phrase (page 11, line 335-336): “HIF-1 is a heterodimeric protein consisting of HIF-1α and HIF-1 α subunits,” Please correct this information because one is α and the other is β.

Response: This has been corrected; thank you for this hint.

  1. The conclusion should be after the discussion section, please change the order.

Response: It is common to put the conclusion after the Discussion section. However, the position of the Conclusions section at the end (after the Methods section) was required by the journal as specified in the journal’s template. Therefore, we cannot change the order.

  1. The authors should add the limitations of the study.

Response: We have added a Limitations subsection at the end of the Discussion section.

Comments on the Quality of English Language

In general, the language english is fine, but there are some grammatical errors and typos in the manuscript.

Round 2

Reviewer 1 Report

I am glad to see that the authors have made the suggested changes. I have no further comments.

NA

Author Response

We thank this reviewer for his/her positive evaluation of our revision.

Reviewer 2 Report

According to statistical analysis section, authors should clarify how they express the data with  Mann-Whitney?  Because for Mann-Whitney it is appropriate that the data be presented as median ± quartile.

The English Language has been improved in the manuscript. 

Author Response

We thank this reviewer for his/her constructive comment on our manuscript.

Reviewer’s comment: According to statistical analysis section, authors should clarify how they express the data with  Mann-Whitney?  Because for Mann-Whitney it is appropriate that the data be presented as median ± quartile.

Response: Most of our data were normally distributed. For non-normally distributed data such as BW change, blood glucose concentration, hematocrit, RV dP/dtmax and min etc., we have replaced the means ± SEM by medians (25th; 75th percentile) (see Tables 1-3).